# Finding Transformer Circuits with Edge Pruning

**Adithya Bhaskar**      **Alexander Wettig**      **Dan Friedman**      **Danqi Chen**
Princeton Language and Intelligence (PLI), Princeton University
`adithyab@princeton.edu`
`{awettig, dfriedman, danqic}cs.@princeton.edu`

## Abstract

The path to interpreting a language model often proceeds via analysis of circuits—sparse computational subgraphs of the model that capture specific aspects of its behavior. Recent work has automated the task of discovering circuits. Yet, these methods have practical limitations, as they rely either on inefficient search algorithms or inaccurate approximations. In this paper, we frame automated circuit discovery as an optimization problem and propose *Edge Pruning* as an effective and scalable solution. Edge Pruning leverages gradient-based pruning techniques, but instead of removing neurons or components, it prunes the *edges* between components. Our method finds circuits in GPT-2 that use less than half the number of edges compared to circuits found by previous methods while being equally faithful to the full model predictions on standard circuit-finding tasks. Edge Pruning is efficient even with as many as 100K examples, outperforming previous methods in speed and producing substantially better circuits. It also perfectly recovers the ground-truth circuits in two models compiled with Tracr. Thanks to its efficiency, we scale Edge Pruning to CodeLlama-13B, a model over $100\times$ the scale that prior methods operate on. We use this setting for a case study comparing the mechanisms behind instruction prompting and in-context learning. We find two circuits with more than $99.96\%$ sparsity that match the performance of the full model and reveal that the mechanisms in the two settings overlap substantially. Our case study shows that Edge Pruning is a practical and scalable tool for interpretability and sheds light on behaviors that only emerge in large models.[1]

## 1   Introduction

Mechanistic interpretability strives to understand models via bottom-up descriptions of their components (e.g., attention heads and MLPs in Transformers [Vaswani et al., 2017]). This typically proceeds via the identification and analysis of a circuit [Olah et al., 2020, Elhage et al., 2021]—a sparse computational subgraph of the model that captures the aspects of its behavior we wish to study. The arduous process of identifying circuits (e.g., Wang et al. [2023]) was recently automated by ACDC [Conmy et al., 2023] and EAP [Syed et al., 2023]. However, ACDC uses an expensive greedy search that ablates each edge to estimate its importance. It cannot scale to datasets beyond a few hundred examples or to billion-parameter models. EAP, on the other hand, uses gradient-based linear approximations of activation patching to estimate the importance of all edges simultaneously. While fast, these first-order approximations often sacrifice faithfulness to the full model. Besides, this approach ignores the impact of the presence/absence of other edges on the score.

In this paper, we frame circuit discovery as an optimization problem and tackle it via gradient-based pruning, rather than discrete search or first-order approximations. As such, we adapt pruning for the goal of circuit discovery instead of model compression. Rather than components, we prune the

---

[1]We release our code and data publicly at `https://github.com/princeton-nlp/Edge-Pruning`.

38th Conference on Neural Information Processing Systems (NeurIPS 2024).

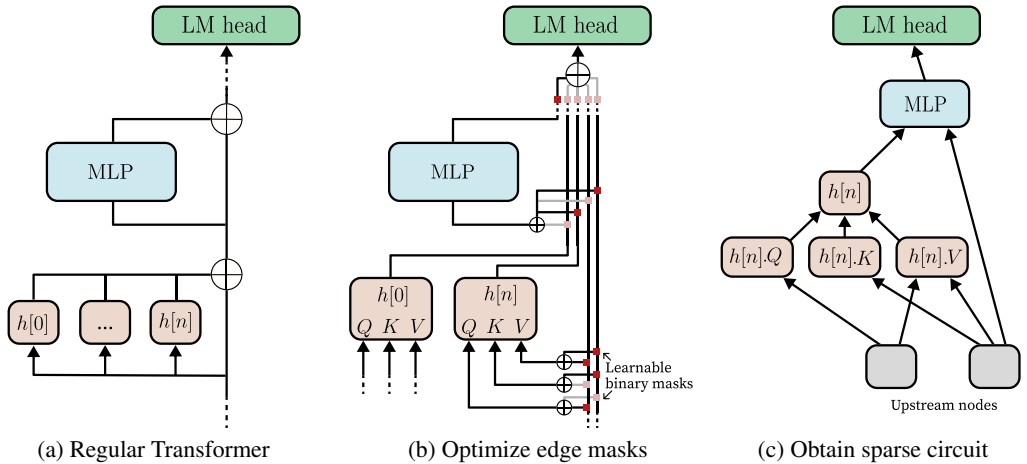

(a) Regular Transformer      (b) Optimize edge masks      (c) Obtain sparse circuit

Figure 1: Edge Pruning disentangles the residual stream and optimizes continuous masks on the read operations via gradient descent. Discretizing the masks to $\{0, 1\}$ yields the final circuit. The full model corresponds to the case where all masks equal 1.

edges between components and replace missing edges with counterfactual activations from corrupted examples. We enable this by replacing the residual stream of a Transformer (Figure 1a) with a *disentangled residual stream* [Lindner et al., 2023, Friedman et al., 2023], which retains a list of all previous activations. This allows us to introduce edge masks that determine from which components to read. We then leverage discrete optimization techniques such as $L_0$ regularization [Louizos et al., 2018] to optimize these edge masks and produce sparse circuits (Figure 1c).

We evaluate our approach, Edge Pruning, on four fronts: (1) we measure how faithfully the discovered circuits describe the behavior of the full model, (2) we verify if it can recover ground-truth circuits in Tracr models [Lindner et al., 2023] compiled from known program descriptions, (3) we evaluate how the method scales to more examples and (4) we assess its ability to find extremely sparse circuits in multi-billion parameter models. On four standard circuit-finding tasks, Edge Pruning finds circuits in GPT-2 Small [Radford et al., 2019] that are consistently more faithful to the full model and have better task performance than circuits found by prior methods. The gap is especially pronounced on more complex tasks like multi-template IOI [Wang et al., 2023], where we find circuits that have $2.65\times$ fewer edges but describe model outputs just as faithfully as the circuit found by the next-best method. We show that Edge Pruning scales effectively to a version of IOI with 100K examples, where it outperforms prior methods in terms of speed and performance. Edge Pruning also perfectly recovers ground-truth circuits in two models compiled from known program descriptions with Tracr.

Finally, we establish that Edge Pruning scales to CodeLlama-13B [Rozière et al., 2024]—100× the size of models typically tackled by automated circuit discovery methods—in a case study. Specifically, we compare the mechanisms behind instruction-prompting and in-context learning [Brown et al., 2020] on Boolean Expressions—a task adapted from the BBH [Suzgun et al., 2022] benchmark. Edge Pruning finds circuits with just 0.04% of model edges that match the model's performance in either setting. Interestingly, the few-shot circuit performs well when instruction-prompted, and vice versa. The two circuits also have a substantial overlap (62.7% edges of the sparser circuit), and the circuit formed by this intersection also performs significantly above chance on the task. We infer that the model relies on shared mechanisms in the two settings. This case study demonstrates how Edge Pruning can inform the analysis of phenomena that only emerge in large models.

In summary, our contributions are as follows:

1. We propose Edge Pruning, an effective and scalable method for automated circuit finding.

2. We demonstrate that Edge Pruning is competitive with or better than state-of-the-art methods on simple tasks, and significantly superior on more complex ones, in terms of faithfulness and performance. Edge Pruning also scales well with more examples. Further, it perfectly recovers ground-truth circuits in two Transformers compiled by Tracr.

3. We scale Edge Pruning to CodeLlama-13B—a model over $100\times$ larger than GPT-2 Small—on a task adapted from BBH. Our case study finds that mechanisms underlying in-context learning and instruction-prompting in CodeLlama-13B for this task overlap significantly.

## 2  Background: Circuit Discovery

The goal of circuit discovery is to facilitate a mechanistic understanding of Transformers by identifying the subset of a model's computational graph that is most relevant to a particular model behavior. In this section, we define the computational graph of a Transformer, formalize the objective for circuit discovery, and discuss the approaches of previous work.

**The computational graph of Transformers.**  The Transformer architecture consists of a sequence of layers, namely attention layers and MLPs, which operate on the *residual stream* (Figure 1a) [Elhage et al., 2021]. The $i$'th layer $f_i$ reads the current state of the residual stream $h_i$, computes its activations $y_i = f_i(h_i)$, and applies it as an additive update to the residual stream $h_{i+1} = h_i + y_i$. We can expand this recurrence to make the dependence on prior outputs explicit:

$$y_i = f_i \left( y_0 + \sum_{j=1}^{i-1} y_j \right), \tag{1}$$

where $y_0$ is the initialization of the residual stream with the input embeddings. We can represent the dependencies between layers as directed edges in a *computational graph*, where the edge $j \rightarrow i$ denotes the connection between the output of layer $j$ to the input of layer $i$. Note that the computational graph may be defined at a more granular level. For instance, Conmy et al. [2023] split attention layers into multiple parallel attention heads, and represents each head by four interconnected nodes. The query/key/value nodes receive separate input edges from previous layers, and the output node has outbound edges to downstream layers. We also follow this convention.

**Circuits as subgraphs.**  A circuit is a computational subgraph $\mathcal{C} \subset \mathcal{G}$, where $\mathcal{C}$ and $\mathcal{G}$ denote the set of edges in the circuit and full model, respectively [Olah et al., 2020]. How do we model a Transformer with a missing edge $j \rightarrow i$? Instead of simply removing the term $y_i$ from the sum of inputs to node $i$, we adopt the approach of *interchange ablation* [Geiger et al., 2020, Zhang and Nanda, 2024]. For each example $x$, the user provides a corrupted example $\tilde{x}$, which should consist of a small change to $x$ that would result in a different label in the task. We use $\tilde{x}$ as input to the full model to compute corrupted activations $\tilde{y}_j$ for all nodes. When an edge $j \rightarrow i$ is removed from a circuit, we replace the contribution of $y_j$ at the input of node $i$ with the corrupted activation $\tilde{y}_j$. This ensures that the summed activations remain in-distribution [Zhang and Nanda, 2024] and it frames the decision to remove an edge as a counterfactual intervention [Vig et al., 2020].

**Circuit discovery.**  The goal of circuit discovery [Olah et al., 2020] is to find a sparse subgraph that describes the behavior of the full model on a particular task. We use $p_{\mathcal{C}}(y \mid x, \tilde{x})$ to denote the output of the Transformer circuit given original and corrupted examples $x, \tilde{x}$, and denote the output of the full model as $p_{\mathcal{G}}(y \mid x)$ as the output of the full model. Formally, circuit discovery has the objective,

$$\arg \min_{\mathcal{C}} \mathbb{E}_{(x,\tilde{x}) \in \mathcal{T}} \left[ D(p_{\mathcal{G}}(y \mid x) \mid\mid p_{\mathcal{C}}(y \mid x, \tilde{x})) \right], \quad \text{subject to } 1 - |\mathcal{C}|/|\mathcal{G}| \geq c \tag{2}$$

where the constraint enforces a target sparsity of the circuit. $\mathcal{T}$ denotes the task distribution of interest, for which the user curates pairs of clean and corrupted examples $(x, \tilde{x})$ that differ in crucial task features. The loss function $D$ should capture the discrepancy between the outputs of the full model and the circuit; for language models, a natural choice is the KL divergence between token predictions.

**Previous approaches.**  We now discuss how previous methods approximate this combinatorial optimization problem and the limitations of their approaches.

1. **ACDC** [Conmy et al., 2023] proposes to solve the above objective using *greedy search*—at each iteration, ACDC evaluates the effect of removing each edge individually, and removes any edge whose effect on the target metric is less than a specified threshold. This fails to capture the relative importance of edges and their interaction. Furthermore, the number of steps of the algorithm scales linearly with the number of edges, which is prohibitive at larger model sizes (e.g., CodeLlama-13B with 3.88M edges).

2. **Edge Attribution Patching (EAP)** [Syed et al., 2023] makes a *linear (first-order) approximation* of activation patching to assign an importance score to each edge. This defines a ranking over edges, from which the top-$k$ edges are used to form a circuit of a specific sparsity. While the linear approximation can compute the importance scores efficiently in a single step, it is likely to find suboptimal solutions to the circuit discovery problem.

3. Conmy et al. [2023] compare to two **pruning-based approaches**. These either (1) prune attention heads based on estimated importance scores [Michel et al., 2019], or (2) perform structured pruning of nodes to identify the most important nodes [Cao et al., 2021]. These approaches perform worse than ACDC [Conmy et al., 2023]. Our approach differs in that we prune edges instead of neurons or nodes. This allows us to optimize at a finer granularity but introduces an additional challenge as we will discuss in Section 3.

## 3 Method: Edge Pruning

In *structured pruning* [Wang et al., 2020, Xia et al., 2022], components such as layers and attention heads are removed to increase the inference efficiency of models. The removal of a component can be modeled by a binary mask, which is relaxed to a continuous parameter to be trainable with gradient-based optimization. While structured pruning produces subgraphs with fewer nodes, they are typically too coarse-grained to help with the mechanistic interpretability of a model's computations.

We propose Edge Pruning, where we define masks not over nodes but over the *edges* connecting them. Specifically, we freeze the original model weights and introduce new trainable parameters $\boldsymbol{z} \in [0, 1]^{|\mathcal{G}|}$, where $|\mathcal{G}|$ is the number of edges in the Transformer, and the parameter $z_{ji}$ is a relaxed binary mask for the edge $j \to i$. In other words, the pruning mask indicates whether an edge is included ($z_{ji} = 1$) or removed ($z_{ji} = 0$) from the computational graph of a circuit. This formulation allows us to find subgraphs with greater granularity and precision compared to structured pruning, as the number of edges scales quadratically with the number of nodes in a model's computational graph.

While structured pruning discards pruned nodes by setting their activation to 0, the application to interpretability calls for more careful treatment of missing nodes and edges. Specifically, the activation of a removed edge $j \to i$ should be replaced by the interchange activation obtained from the corrupted version of the example (Section 2). To allow gradient-based optimization, we model the process as the masks continuously interpolating between the clean and corrupted activation. Specifically, we parameterize the $i$'th component as,

$$
y_i = f_i \left( z_{0i} y_0 + (1 - z_{0i}) \tilde{y}_0 + \sum_{\substack{1 \leq j < i \\ j \text{ upstream of } i}} \left( z_{ji} y_j + (1 - z_{ji}) \tilde{y}_j \right) \right), \tag{3}
$$

where $\{\tilde{y}_j\}$ denote the corrupted activations corresponding to $\tilde{x}$.

Our formulation has a key challenge. Each node sees a different combination of activations depending on incoming edges, and thus a different residual stream. Thus, we can no longer add the activations $y_i$ immediately to the residual stream, i.e. $h_{i+1} = h_i + y_i$, as shown in Figure 1a. Instead, we modify the Transformer architecture to retain a so-called *disentangled* residual stream [Friedman et al., 2023], in which the activations $y_i$ are concatenated to a list of all previous activations $(y_0, y_1, \ldots, y_{i-1})$. Then, we dynamically aggregate these activations at the input of each node (Equation 3 and Figure 1b).

In practice, concatenation increases the GPU memory footprint during training compared to regular structured pruning (Appendix A), but it is necessary for optimizing over edges between nodes that are separated by many layers. Despite the memory overhead, we demonstrate in Section 5 that we can scale our method to large models by parallelizing training over multiple GPUs.

We directly optimize the objective in (2) by performing stochastic gradient descent with respect to the edge weights $\boldsymbol{z}$. The target sparsity is enforced via $L_0$ regularization with a Lagrangian term. We leverage the formulation of Louizos et al. [2018] to model the masks as hard concrete parameters and to circumvent the non-differentiability of the L0 term. At the end of the training, the edge weights are converted to binary masks based on a threshold (e.g., 0.5), which uniquely determines the produced circuit (Figure 1c). We now describe this process in more detail.

**Details of the Edge Pruning process**    Our formulation of pruning is based on that used by CoFi Pruning [Xia et al., 2022]. Specifically, we model the masks $z$ based on the hard concrete distribution as done by Louizos et al. [2018]:

$$\mathbf{u} \sim \text{Uniform}(\epsilon, 1 - \epsilon)$$
$$\mathbf{s} = \sigma\left(\frac{1}{\beta} \cdot \frac{\mathbf{u}}{1 - \mathbf{u}} + \log \boldsymbol{\alpha}\right)$$
$$\tilde{\mathbf{s}} = \mathbf{s} \times (r - l) + l$$
$$\mathbf{z} = \min(1, \max(0, \tilde{\mathbf{s}}))$$

where $\sigma$ refers to the sigmoid function, $\epsilon = 10^{-6}$, and $\log \boldsymbol{\alpha}$ indicates that the logarithm is applied element-wise. We fix the temperature $\frac{1}{\beta} = \frac{2}{3}$. The last two lines stretch the distribution to $[l, r] = [-0.1, 1.1]$ and accumulate the "excess" probability on either side to 0 and 1, respectively. The log alphas $\log \boldsymbol{\alpha}$ are the learnable parameters in this formulation.

Following, Wang et al. [2020], a target sparsity is enforced via a Lagrangian term [Louizos et al., 2018]. If the current sparsity is $s$, the term, parametrized by a reference value $t$ is

$$\mathcal{L}_s = \lambda_1 \cdot (t - s) + \lambda_2 \cdot (t - s)^2$$

$\lambda_1$ and $\lambda_2$ are also updated during training via gradient *ascent* to keep the regularization tight. We vary the value of $t$ throughout training, linearly increasing it from 0 to a target value, as outlined in Appendix A. Although it may be useful to think of $t$ as a "target" sparsity, it is only a number. The runs usually converge to a value slightly below $t$, so it is prudent to set it to a value *greater than* 1—although $s$ can then never reach the target value, it will be pushed to higher sparsities.

We have two sets of masks $z$. The first set associates a $0 - 1$ value $z_e$ with each edge $e \equiv (n_1, n_2)$ in the computational graph. The second set tags each *node* of the graph $n$ with a $0 - 1$ value $z_n$. The latter specifies whether a node is "active", i.e., producing output. In effect, the presence of an edge $e \equiv (n_1, n_2)$ is determined by the binary mask

$$\tilde{z}_{(n_1, n_2)} = z_{(n_1, n_2)} \times z_{n_1}$$

We initially only used edge masks but found that the method would have difficulty converging to high sparsities (i.e., end up at low sparsities). Introducing a second set of masks allows the process to eliminate many edges quickly, accelerating the removal of unimportant components. However, the lagrangian above only applies to the edge masks. This is fine since the node masks can only remove further edges, not introduce new ones on top of those chosen by the edge masks. The final loss is

$$\mathcal{L} = \mathcal{L}_{\text{KL}} + \mathcal{L}_{\text{edge},s}$$

## 4    Experiments

### 4.1    Experimental Setup

**Methods.**    We compare **Edge Pruning** with a KL loss to **ACDC** and **EAP** in our experiments. Both are outlined in Section 2. We do not compare to other pruning-based methods, as Conmy et al. [2023] found them to perform much worse than ACDC. We list the hyperparameters used in Appendix A. The experiments in this section are all performed on GPT-2 Small (117M).

**Tasks.**    Prior works evaluate their methods on the same examples used to find circuits. In a departure from this convention, we separate each dataset into `train`, `validation`, and `test` splits, to avoid artifacts caused by overfitting. We use the following tasks.

- **Indirect Object Identification (IOI-t1 and IOI)** [Wang et al., 2023] is a task with instances of the format "*Friends Juana and Kristi found a mango at the bar. Kristi gave it to →* *Juana*". Conmy et al. [2023] use a version with a single template, which we refer to as **IOI-t1**—this version has 50 examples in each split. We also compare the methods on a variant (**IOI**) with 30 templates found on HuggingFace[2]. We randomly select 200 examples each for the `train` and `validation` splits, and $36,084$ examples for the `test` split.

---

[2]https://huggingface.co/datasets/fahamu/ioi/; an example template is "*Then, B and A had a long argument. Afterwards B said to → A*".

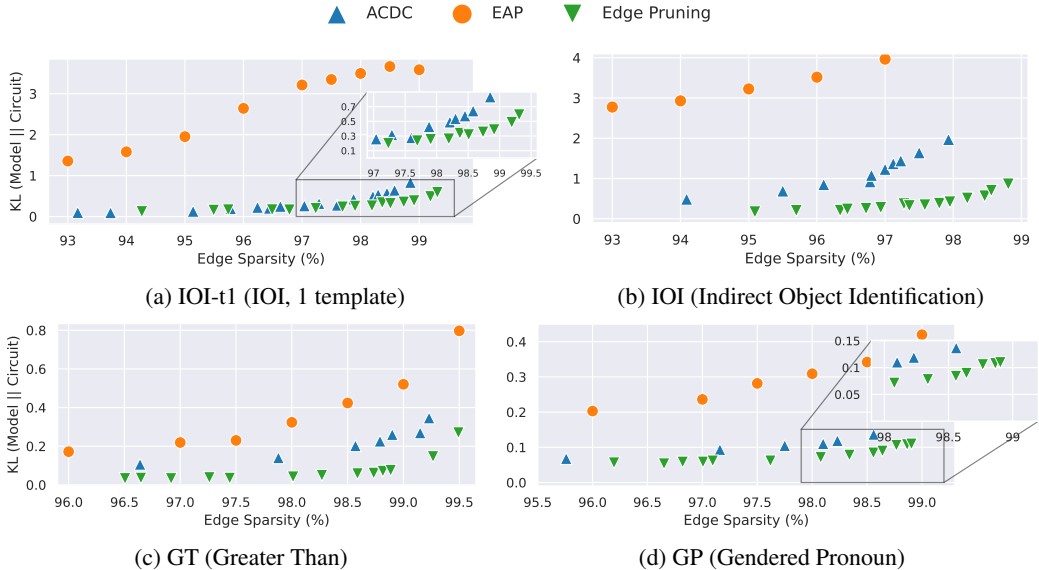

Figure 2: The faithfulness of the methods, given the KL divergence between the model and obtained circuits (*lower is better*). On IOI-t1 and GP, Edge Pruning is competitive at low sparsities and better at high sparsities. It outperforms both ACDC and EAP by a significant margin on IOI and GT.

- **Greater Than (GT)** [Hanna et al., 2023] consists of examples of the format "*The war lasted from the year* 1743 *to* 17 → $xy$". The objective of the task is to place a greater probability on the continuations $44, 45, \ldots, 99$ than $00, 01, \ldots, 42$. Our dataset spans 5 templates, 120 choices for nouns, and the years 1100 through 2199. It has 150 examples in the `train` and `validation` splits, and $12,240$ examples in the `test` split.

- **Gendered Pronoun (GP)** [Athwin et al., 2023] consists of statements of the form "So Evan is a really great friend, isn't → he". We use the templates from the original Colab notebook used by Athwin et al. [2023], but generate more examples as they only work with 5. We use the top $1,000$ most popular baby names for boys and girls each in the year $2000$[3] to generate a dataset with 150 `train` and `validation` examples each, and 378 test examples.

- **Tracr** [Lindner et al., 2023] compiles programs written in the RASP [Weiss et al., 2021] programming language into few-layer Transformers. We evaluate Edge Pruning on how well it recovers ground-truth circuits for two Tracr programs—`xproportion` (proportion of `x`'s in the prefix) and `reverse` (reversing a list). Both tasks were discussed in Weiss et al. [2021] and used by Conmy et al. [2023] in their evaluation.

**Evaluation.** A circuit is faithful to model behavior on a task if we can corrupt all model edges outside the circuit while retaining the model's outputs [Hanna et al., 2024]. We corrupt non-circuit edges with interchange ablation and evaluate the methods' faithfulness as the **KL divergence** between model and circuit outputs. Specifically, we corrupt an example by swapping the placeholder value in the same template with a random example from the dataset. We appraise the circuits' performance on IOI-t1, IOI, and GP via the **Logit Difference** $\log P(\text{correct}) - \log P(\text{misleading})$ between the correct and misleading name/pronoun. For GT, we evaluate the **Probability Difference** $P(yy + 1 : 99) - P(00 : yy - 1)$ between the correct and incorrect ranges. All metrics on GT work with predictions restricted to the set $\{00, 01, \ldots, 99\}$. We always take unrestricted predictions over the entire model vocabulary on other tasks. All non-Tracr experiments use a GPT-2 Small model. Appendix B evaluates additional metrics—including circuit overlap with manually found circuits.

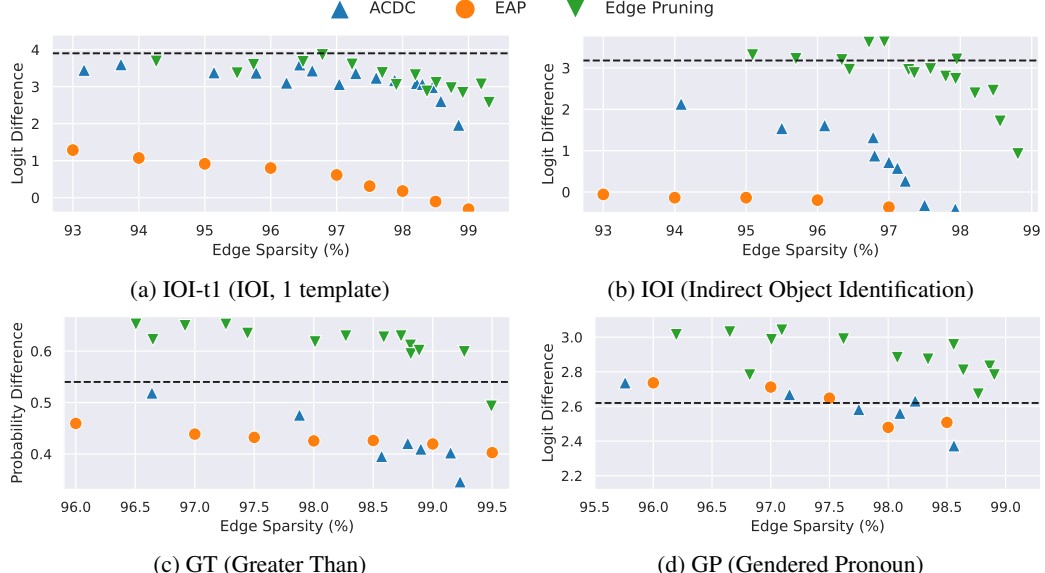

Figure 3: Comparison of circuit performance between methods. We report the Logit Difference $\log P(\text{correct}) - \log P(\text{misleading})$ for IOI-t1, IOI and GP, and the probability difference $P(yy + 1 : 99) - P(00 : yy - 1)$ for GT. Higher is better for all plots. Edge Pruning finds better-performing circuits on all four tasks. The dashed line indicates the performance of the full model.

Table 1: Scaling to a larger IOI dataset: ACDC improves with more examples but its runtime scales prohibitively. EAP is fast but cannot perform as well. Edge Pruning scales effectively to 100K examples, where it is the fastest and most faithful method. All runs use one NVIDIA H100 GPU.

| Method | Sparsity (%) ↑ | 200 examples | | 400 examples | | 100K examples | |
|---|---|---|---|---|---|---|---|
| | | KL ↓ | Time (s) ↓ | KL ↓ | Time (s) ↓ | KL ↓ | Time (s) ↓ |
| ACDC | 96.6 ± 0.1 | 0.92 | 18,783 | 0.88 | 40,759 | - | - |
| EAP | 96.6 ± 0.1 | 3.47 | **21** | 3.66 | **43** | 3.78 | 12,260 |
| Edge Pruning | 96.6 ± 0.1 | **0.25** | 2,756 | **0.22** | 2,931 | **0.20** | **3,042** |

## 4.2 Results

This section compares the three methods on our primary faithfulness and performance metrics. We report additional metrics in Appendix B, and Appendix F shows some circuits found by Edge Pruning.

**Edge Pruning outperforms prior methods on more complex tasks.** Edge Pruning is competitive on IOI-t1 and GP in terms of faithfulness at low sparsities, and slightly better at higher sparsities (Figure 2). It is considerably more faithful on IOI and GT than both ACDC and EAP, especially at higher sparsities. In particular, ACDC does worse than randomly choosing between the two names (KL divergence 0.69) at high sparsities on IOI, whereas Edge Pruning remains better. We hypothesize that the relative simplicity of IOI-t1 and GP—one template or small output space (he/she)—renders local (ACDC) or first-order (EAP) approximations good proxies, potentially explaining the edge of Edge Pruning on IOI and GT. A similar trend is seen in performance (Figure 3): Edge Pruning finds better-performing circuits on all four tasks. Specifically, on IOI, Edge Pruning finds a circuit of $98.8\%$ sparsity that is as faithful and performs as well as the one found by ACDC at $96.8\%$ sparsity—using over $2.65\times$ fewer edges. Interestingly, EAP scales better to higher sparsities than ACDC on GT, delivering respectable performance even at $99.5\%$ sparsity.

**Edge Pruning can scale to 100K examples.** We investigate how the methods scale to more examples at representative sparsities. To this end, we create a large version of the IOI dataset's train

---

[3]https://github.com/aruljohn/popular-baby-names/

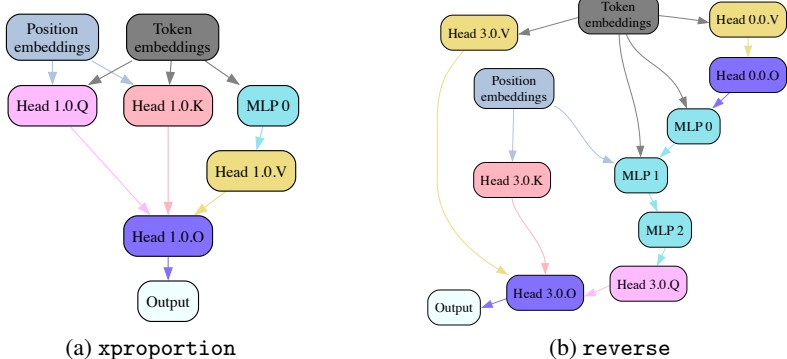

(a) `xproportion`                    (b) `reverse`

Figure 4: The canonical ground-truth circuits for the Tracr-compiled `xproportion` and `reverse` programs. Edge Pruning recovers both circuits perfectly.

split with 100K examples. We hold the number of gradient descent steps for Edge Pruning fixed (Appendix A). Although its runtime would scale linearly with more epochs, at 100K examples all approaches see almost all examples once.[4] Thus, the time reported in Table 1 represents the relative overhead of each method. ACDC shows clear improvements with more examples, but cannot scale well due to prohibitive runtime. EAP, on the other hand, is fast even with more examples. However, it underperforms the other two methods significantly. Edge Pruning efficiently uses more examples and demonstrates both the least runtime and the highest faithfulness by far with 100k examples. We therefore conclude that Edge Pruning is a good fit for complex or mixture distributions where more examples may be needed to specify model behavior.

**Edge Pruning finds ground-truth circuits in Tracr programs.**    To check if Edge Pruning can find the ground-truth circuits, we use Tracr [Lindner et al., 2023] to compile two example programs—`xproportion` and `reverse`—as Transformers. The former yields a 2-layer Transformer that outputs, at each position, the fraction of x's seen so far. The latter yields a 3-layer Transformer that can reverse lists. We use zero ablation following Conmy et al. [2023] (more details in Appendix A). Edge Pruning achieves perfect reconstruction of both circuits (Figure 4).

**Edge Pruning is robust to variance in random initialization**    Appendix D finds that both the resulting sparsity and the faithfulness of the circuits found by Edge Pruning are remarkably consistent across different random initializations of masks. We also investigate there the question of whether multiple different circuits can exist for a given task, and if Edge Pruning can find them.

## 5    Case Study: Scaling to 13B Parameters

We have seen that Edge Pruning can scale efficiently with more examples. We next investigate if it can scale with *model size*. This is increasingly important, given the recent interest in interpreting multi-billion parameter models [Lieberum et al., 2023, Prakash et al., 2024]. Current methods used to interpret such models, while undeniably indispensable, have limitations: path patching [Goldowsky-Dill et al., 2023] identifies important subsets of components but falls short of producing edge-level circuits. Distributed Alignment Search [Geiger et al., 2024, Wu et al., 2023] can verify proposed symbolic execution graphs and align them with the model but requires prior knowledge of the correct symbolic graph, which is nontrivial to obtain.

On the other hand, pruning can scale to large models using model parallelism [Xia et al., 2024]. We thus apply Edge Pruning to a case study on CodeLlama-13B [Rozière et al., 2024]—a model over $100\times$ larger than GPT-2—with a real task. We are inspired by Prakash et al. [2024], who compare base and fine-tuned LMs and find that finetuning enhances existing mechanisms. Instead of comparing base and fine-tuned models, we compare mechanisms in the *same* model with different prompting schemes. Specifically, *we ask whether the same mechanisms underlie (zero-shot) instruction prompted*

---

[4]With our hyperparameters, Edge Pruning sees 96k unique examples (can be higher with more GD steps).

Table 2: Edge pruning finds circuits with 0.03-0.04% of the edges in CodeLlama-13B that match the performance of the full model. The circuits perform well in cross-evaluation and overlap highly, hinting that the same mechanisms explain large parts of instruction-prompted and few-shot behavior.

| Circuit | Num. edges ↓ | Accuracy (%) ↑ | | Exact Match (%) ↑ | |
|---|---|---|---|---|---|
| | | Instr. prompted | Fewshot | Instr. prompted | Fewshot |
| Full model | 3872820 | 82.00 | 89.25 | 100.00 | 100.00 |
| Instruction prompt (IP) | 1041 | 79.25 | 74.50 | 90.00 | 79.00 |
| Fewshot (FS) | 1464 | 75.75 | 87.25 | 84.50 | 91.25 |
| IP ∩ FS | 653 | 72.50 | 68.25 | 79.75 | 72.50 |

*and few-shot behavior* for the task-model pair we study. This case study serves a dual purpose. It demonstrates the scalability of Edge Pruning as a method. It also illustrates how circuit-finding methods may fit into the interpretability arsenal. We are interested in three research questions: (RQ1) Can Edge Pruning find edge-sparse circuits in a 13B model? (RQ2) To what extent do the circuits for instruction and few-shot prompting share the same edges? (RQ3) Does the instruction-prompted circuit perform well when used in a few-shot manner, and vice versa?

**Task and model setup.** We work with the task *Boolean Expressions* from the BBH [Suzgun et al., 2022] benchmark suite. This task consists of instances of the form "*((not False) and False) or (False and True) is → False*". The original dataset only has 250 examples, so we programmatically generate an in-house version of the task. Our dataset has 3840, 767, and 3070 examples in the train, validation, and test splits respectively. Each instance has between 3 and 6 literals, with a maximum nesting depth of 3 and at most 2 consecutive *not*s. We use 3 demonstrations for the few-shot setting. The prompts used for the instruction-prompted and few-shot settings are provided in Appendix E. Our model is the instruction-finetuned version of CodeLlama-13B.[5] It achieves accuracies of 82% and 89.25% in the instruction-prompted (IP) and few-shot (FS) settings, respectively.

**(RQ1) Edge Pruning produces extremely sparse circuits.** We next apply Edge Pruning to the described settings. We isolate one circuit when instruction prompting and one with the few-shot prompt (hyperparameters in Appendix A, which also highlights other optimizations like distributed training and gradient checkpointing). The circuit discovered in the IP setting has $1,041$ edges, corresponding to a $99.97\%$ edge sparsity. That discovered in the FS setting has $1,464$ edges, equivalent to $99.96\%$ edge sparsity. The discovered circuits are evaluated in Table 2. Despite using less than $0.04\%$ of the edges, the circuits closely match the performance of the full model—the few-shot circuit achieves an accuracy of $87.25\%$ and performs within $2\%$ of the full model (when prompted few-shot). The instruction-prompted circuit is accurate within $2.75\%$ of the full model.

**(RQ2) The circuits have a high overlap, and their intersection performs well.** We appraise the intersection of the IP and FS circuits next. The two circuits share 653 edges, accounting for $62.7\%$ of the edges of the sparser (instruction prompted) circuit—this corresponds to an intersection over $1,700\times$ larger than expected by random chance. We further evaluate the circuit formed by this intersection in the instruction prompted and few-shot settings (Table 2). It performs well in the instruction prompted setting, and worse than the model (but still significantly above chance) when prompted few-shot.

**(RQ3) The circuits demonstrate strong performance in cross-evaluation.** We note from Table 2 that the circuit found with few-shot prompting shows strong performance even when instruction prompted. Analogously, the instruction-prompted circuit also performs well in the fewshot setting.

Our case study suggests that the same mechanism (as represented by the intersection above) explains a large part of the performance in both settings—i.e., they do not proceed via disjoint mechanisms. However, the performance gap between the FS and IP ∩ FS circuits is still sizable. Further, we see modest drops in cross-evaluation—e.g., from $87.25\%$ when evaluating the FS circuit few-shot to $75.75\%$ in the instruction prompted setting. This suggests that additional components are needed to

---

[5]https://huggingface.co/codellama/CodeLlama-13b-Instruct-hf

complete the picture. A complete mechanistic description of the components in the two circuits is an exciting avenue for future work, but beyond the scope of this case study.

**Manual analysis of the CodeLlama-13B circuit.**   Interpreting a circuit in such a large model—even if very sparse— remains a challenging task. We isolate a small region of the circuit and identify curious behavior in it in Appendix F, leading to an intriguing conjecture. Nonetheless, we believe that a thorough study requires more analysis, which is beyond the scope of this paper (but makes for exciting future work).

## 6   Related Work

**Circuits.**   By reducing a large model to a sparse subgraph, circuits help interpret internal model computations  [Olah et al., 2020, Elhage et al., 2021], and several visualization tools have been developed to aid this process [Sakarvadia et al., 2023, Katz and Belinkov, 2023, Tufanov et al., 2024]. Circuits were originally found manually [Hanna et al., 2023, Athwin et al., 2023], but this has recently been automated by tools like ACDC [Conmy et al., 2023]. ACDC uses activation patching [Vig et al., 2020] to knock out unimportant edges. Other approaches instead estimate the importance of each edge via attribution scores [Nanda, 2022]; this approach was used by EAP [Syed et al., 2023]. Ferrando and Voita [2024] use attribution patching to identify domain-specific model components in Llama-2-7B. Kramár et al. [2024] note that attribution patching may lead to incorrect approximations, and propose a variant with reduced error. In concurrent work, Hanna et al. [2024] argue that faithfulness metrics are better for evaluating circuits than measuring overlap with manual circuits. Recent work has explored other notions of a circuit. Inspired by the fact that Sparse Autoencoders (SAEs) can find human-interpretable features in LM activations [Cunningham et al., 2023], Marks et al. [2024] find circuits over these features.  Wu et al. [2023] align computation in Alpaca [Taori et al., 2023] with a proposed symbolic algorithm [Geiger et al., 2024]. Our method is orthogonal to these developments.

**Pruning.**   Pruning [LeCun et al., 1989] drops parameters or layers of a language model for space efficiency and potential speedups. *Structured pruning* [Wang et al., 2020, Xia et al., 2022] imposes some regularity on the resulting subnetworks, such as an equal fraction of preserved parameters in each layer. Doing so allows it to achieve substantial speedups on GPU hardware at the cost of lower compression. In contrast, unstructured pruning [LeCun et al., 1989, Hassibi and Stork, 1992] does not impose such constraints. *Channel pruning* [He et al., 2017] is a form of structured pruning that prunes input channels in vision models, which has been adapted for neural architecture search [e.g. Li et al., 2022]. Pruning has occasionally been used as part of an interpretability effort, but mostly at the level of neurons [Michel et al., 2019, Jain et al., 2023], or less commonly, attention heads/MLPs [Cao et al., 2021]. Our work finds circuits by pruning the edges between components instead.

## 7   Conclusions

In this paper, we introduce Edge Pruning to find circuits by pruning edges between components. We find that it discovers sparse, faithful circuits, and we demonstrate its scalability to large datasets and large models. We close by discussing its limitations, and how future work may address them.

**Limitations.**   We acknowledge that with small datasets, approximation-based approaches like EAP are faster than Edge Pruning. Circuit discovery with Edge Pruning may also require more GPU memory than these methods—especially at scale—where we use 32 H100 GPUs for CodeLlama-13B (Appendix A). Future work may precede Edge Pruning with a fast, approximate method like EAP to balance efficiency and performance. We note that even at very high sparsities, circuits for large models can still have hundreds of edges, and their full interpretation remains challenging. Further automating interpretability [Bills et al., 2023] is a compelling avenue for future research. Finally, we note that even with perfect faithfulness to the model outputs, a circuit can misrepresent the necessary computations in the full model, thus leading to interpretability illusion [Makelov et al., 2024]. Better metrics are needed to reveal these possibilities in practice.

**Societal and ethical impact.** Our work aims to facilitate the process of understanding and explaining large foundation models, which is crucial for their continued safe development and deployment. We do not foresee Edge Pruning being used towards adverse societal or ethical ends.

## Acknowledgements

We are thankful to Tianyu Gao, Zirui Wang, and Mengzhou Xia for their helpful discussions regarding the experiments. Input by Abhishek Panigrahi and Carlos Jiminez was also instrumental to this project. We also thank Howard Chen and Tianyu Gao for their help in proofreading and improving the writing in this paper. AB gratefully acknowledges the support of a Hisashi and Masae Kobayashi *67 Fellowship. This research is also funded by the National Science Foundation (IIS-2211779) and a Sloan Research Fellowship.

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

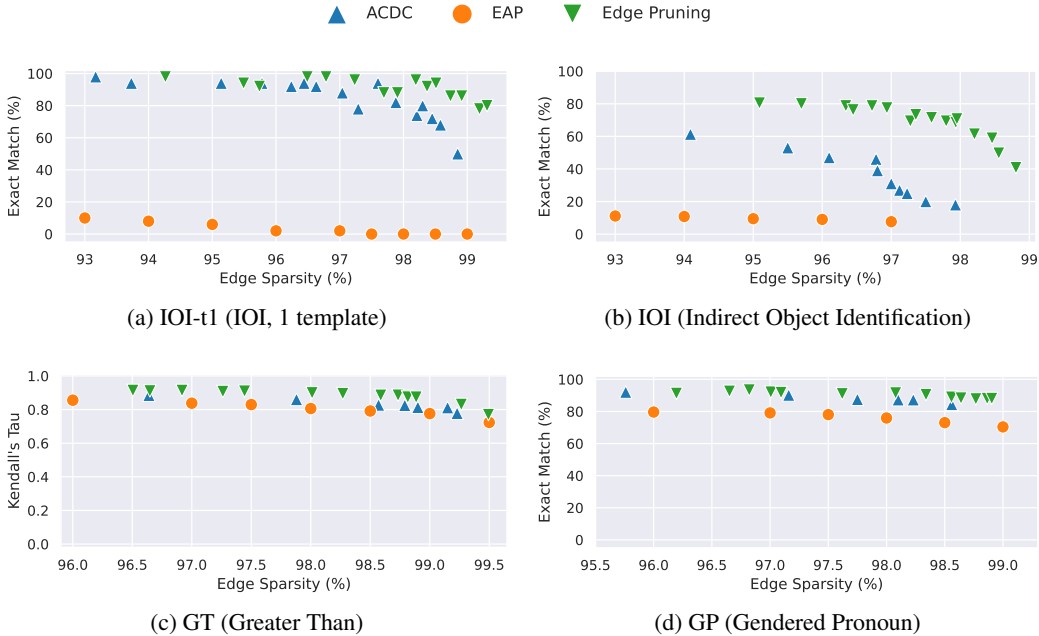

Figure 5: Our secondary metric for measuring faithfulness is the Exact Match percentage between the model and circuit predictions on IOI-t1, IOI, and GP. On GT, we use the Kendall's Tau score between the model and circuit rankings of $00, 01, \ldots, 99$ as the secondary metric. Edge Pruning is the most faithful method on all four tasks, with the difference being especially pronounced for IOI.

## A  Hyperparameters and Computational Details for Edge Pruning

In this appendix, we list the hyperparameters used for the various experiments in the main text of the paper. All of our runs use the Adam [Kingma and Ba, 2015] optimizer with $\epsilon = 10^{-8}$ and $(\beta_1, \beta_2) = (0.9, 0.999)$.

**GPT-2 experiments**  For all tasks, we used a sequence length of $64$ tokens with padding. A batch size of $32$ was adopted, and the learning rate for both the edge and node masks, as well as for the lagrangians $\lambda$ for both, was set to $0.8$. IOI-t1 was an exception: here, we set all the above learning rates to $1$ for all runs. The total number of optimization steps was $3000$, and the target edge and node sparsities were linearly increased starting from $0$ over the first $2500$ steps. Evaluation and checkpointing were performed every $64$ steps but we always used the final checkpoint to report results. To produce the scatterplots, we varied the edge target up to $1.1$ but held the node target largely fixed for each task. These values were $0.72$ for IOI-t1 and IOI, $0.68$ for GT and $0.69$ for GP. These values were chosen based on a small number of pilot runs, and we expect that a grid search can improve results further.

We also wish to make several remarks about our implementation. We turned off dropout for all runs since it made the optimization noisy. Our threshold for the final rounding is not a pre-determined value. Instead, we compute the average value of all entries of all masks, and brand that the desired sparsity. Then, we perform a binary search for a threshold such that the fraction of entries rounded to 1 equals this desired sparsity. The thresholds found this way usually fell between $0.2$ and $0.8$. This also allows the user to achieve exactly the desired sparsity by setting a different threshold. We implement all of our code by implementing modified versions of the HuggingFace model classes, as it allows us to use the HuggingFace Trainer and its optimizations out of the box. Our code also natively supports Flash Attention, though none of our results use it. Finally, we note that the role of $\lambda_1$ in the lagrangian term is to allow (and indeed, encourage), "shooting past" $t$ when optimizing $s$ due to momentum. This prevents the model sparsities from "settling into" a mode where they lag behind the targets by a constant but non-zero amount throughout pruning.

**Tracr experiments**  For both programs, we fix the $\lambda_1$ values to $0$ and only optimize $\lambda_2$, as described in Section 3. For the `xproportion` program, we use an edge target of $0.92$ and a node target of $0.4$. The edge and node mask learning rates were $1$, and that for the lambdas was $0.0001$. A total of $720$ optimization steps were performed with a batch size of $16$, of which $640$ was used for target warmup. The learning rates were warmed up linearly over the first $96$ steps. A sequence length of $5$ was used.

Initially, for `reverse`, setting the regularization learning rate was tricky—it was easy to end up not regularizing enough or overdoing it. Thankfully, an easy remedy was to increase the number of steps to $6000$ (of which the first $5900$ warmed up the edge and node targets, and the first $1500$ warmed up the learning rates). This allowed us to set a relatively higher learning rate for the lambdas ($0.001$), along with an aggressive edge target of $1.02$. The node target was set to $0.1$. The learning rates of the log alphas and lambdas were $0.03$ and $0.001$, respectively. Despite using $6000$ steps, the run took under $5$ minutes on one NVIDIA A100.

**CodeLLama-13B experiments**  For our CodeLlama-13B experiments, we use a learning rate of $0.8$ for both the edge masks and the node masks. In a departure from the choice of Section 3, we also include a separate lagrangian term over node masks:

$$\mathcal{L}_{\text{node},s} = \lambda_{1,\text{node}} \cdot (t_{\text{node}} - s_{\text{node}}) + \lambda_{2,\text{node}} \cdot (t_{\text{node}} - s_{\text{node}})^2$$

The reason for this choice was that, in our preliminary runs with small Sheared Llama [Xia et al., 2024], we found that this would achieve higher sparsities. We use a learning rate of $0.4$ for all of the lambdas. The target edge and node sparsities are set to $1.2$ and $0.7$, respectively. We use $6000$ steps with a batch size of $1$. The first $200$ steps linearly warm up the learning rate, while the target sparsities are linearly increased over the first $5500$ steps. We enable gradient checkpointing, as well as FSDP [Zhao et al., 2023] with full sharding in BF16 precision. The maximum sequence lengths for the instruction-prompted and few-shot settings were $64$ and $72$, respectively.

We also comment here on the computational resources used for the runs.

**Computational details.** The Tracr experiments use one NVIDIA A100 with 80 GB of memory. The GPT-2 experiments use either one NVIDIA A100 or one H100 (both 80 GB) each. The experiments of Table 1 all use one NVIDIA H100 for a fair runtime comparison. Each CodeLlama-13B run utilizes 32 H100 GPUs and 600 gigabytes of CPU memory. The typical runtime of a GPT-2 pruning run was about 45 minutes, and that of a Tracr run was under 5 minutes. The CodeLlama runs each took around 35 hours. We estimate the total computational budget to be around 5000 GPU hours.

# B   More results

We show more results on faithfulness and performance metrics in this appendix. Specifically, we evaluate on one alternate faithfulness (agreement) metric and one additional performance metric. For the former, we choose **Exact Match** percentage as the agreement metric on IOI-t1, IOI and GP. For GT, we instead report the **Kendall's Tau** score over the rankings of $00, 01, 02, \ldots, 99$ as induced by the output logits of the model and circuit, which is then averaged across examples. Figure 5 plot these metrics for the three approaches. We see that Edge Pruning is consistently the most faithful method on all four tasks, with the gap to the next-best method being large for IOI.

Our choice of the performance metric is **Accuracy** for IOI-t1, IOI and GP. For GT, we instead compute a variant of Probability Difference called Probability Difference 10, given by $P(yy + 1 : yy + 10) - P(yy - 10 : yy - 1)$. Note that the original probability difference, $P(yy + 1 : 99) - P(00 : yy - 1)$ can be gamed by always predicting $99$. The new variant overcomes this obstacle by measuring the sharpness of the cutoff. The results, shown in Figure 6, echo the results of the main text: edge pruning is competitive on GP, and outperforms the other methods in IOI-t1, IOI and GT.

We also compare Edge Pruning to ACDC in terms of circuit overlap with manually reverse-engineered circuits. Since the manual circuits only identified important components and not the edges between them, we plot node (component) ROC curves in Figure 7, where we consider a node included in a circuit if at least one edge incident to it is included. Note that the IOI manual circuit only studied attention heads, so we ignore MLP nodes in the corresponding circuits. The results show that Edge Pruning is competitive with ACDC on circuit overlap metrics. Nevertheless, we emphasize that manually reverse-engineered circuits are not guaranteed to be optimal since they also investigate one ablation at a time without considering interactions between ablations. As such, we echo Hanna et al. [2024]'s suggestions of using circuit faithfulness metrics over circuit overlap.

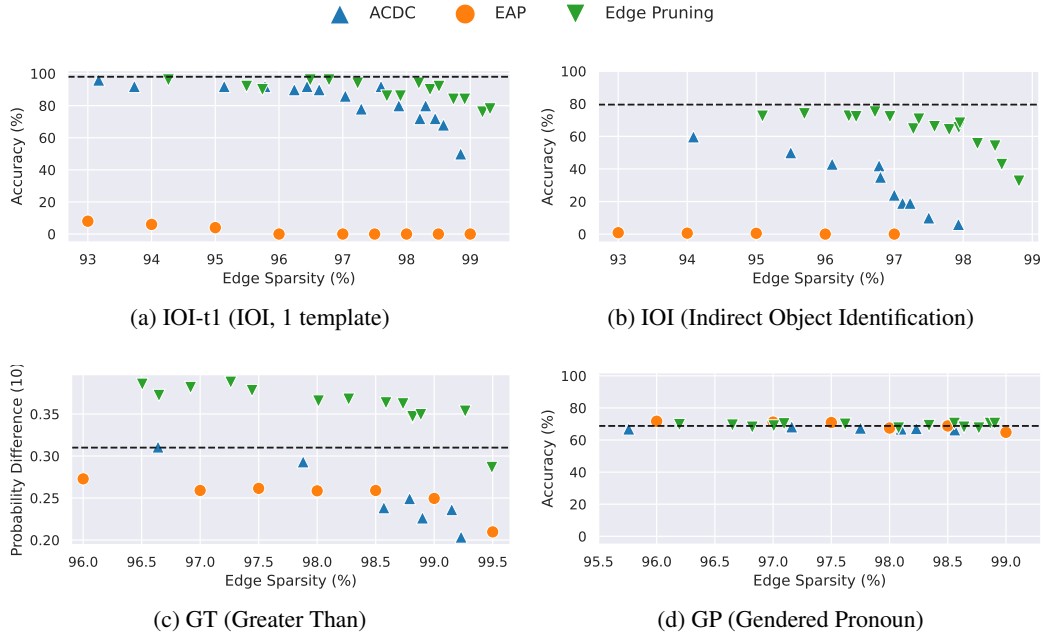

Figure 6: Comparison of the various methods on our secondary performance metric—accuracy in the case of IOI-t1, IOI and GP, and Probability Difference 10 for GT (given by $P(yy + 1 : yy + 10) - P(yy - 10 : yy - 1)$). Once again, Edge Pruning is competitive on GP, and outperforms other methods on IOI-t1, IOI and GT. The dashed lines indicate full model performance.

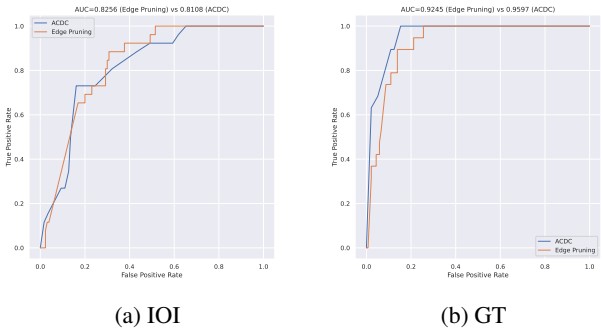

(a) IOI  (b) GT

Figure 7: ROC curves against manual circuits for Edge Pruning and ACDC. The AUC is slightly higher for Edge Pruning on IOI, and slightly lower on GT.

## C  Edge Faithfulness

In other sections and appendices, we have taken up the *output faithfulness* of Edge Pruning, i.e., whether the output distribution of the circuits matches that of the model. Here, we consider another *edge faithfulness*—an edge important for the model should also be important for the circuit. Concretely, given a circuit $C$ of a model $M$, we measure for each edge $e \in C$, $m_e \equiv \text{KL}(M || M \setminus \{e\})$ and $c_e \equiv \text{KL}(M || C \setminus \{e\})$, i.e., how much removing the edge from the circuit or model affects its output distribution. For a method to be faithful, we expect to see a strong positive correlation between the two values, especially for edges where $m_e$ is large. We plot the two values against each other on the four tasks for four representative circuits found by Edge Pruning in Figure 8. The figure also provides the sparsities of each circuit. On all four tasks, whenever an edge is important to the model, it is also important to the circuit. Thus, studying the circuit to infer the role/importance of the components is a good proxy for the full model. On the other hand, we note that some edges are completely unimportant for the model, but ablating which perturbs the circuit KL by a small

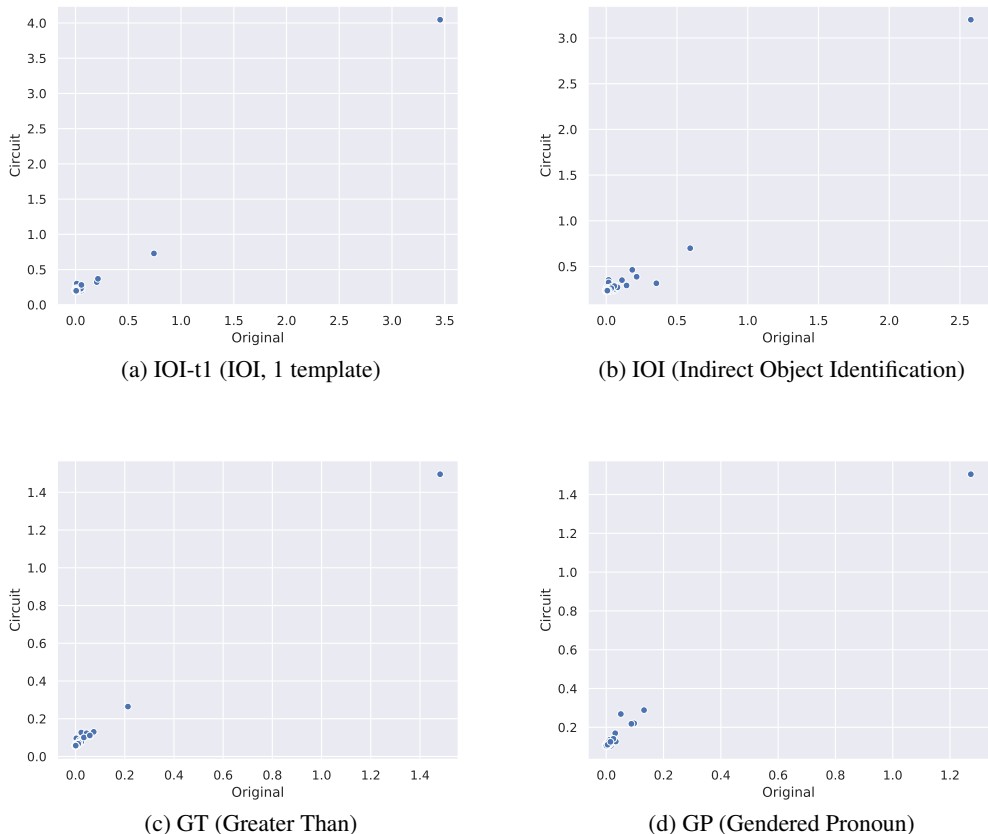

(a) IOI-t1 (IOI, 1 template)

(b) IOI (Indirect Object Identification)

(c) GT (Greater Than)

(d) GP (Gendered Pronoun)

Figure 8: The KL divergences of the model and circuit, upon ablating individual circuit edges from each, measured against the full model. We see that all components important to the model are also important to the circuits, with an almost linear correlation between the two quantities. The circuits shown here have sparsities of $97.23\%, 96.44\%, 98.59\%$, and $98.77\%$, respectively.

amount. This perturbation is much smaller than the ones seen in the former case above, but still non-negligible. This is not surprising, as circuit-finding methods may miss backup components that are deemed unnecessary for performance, and therefore be more sensitive to edge ablations. Alternatively, models may display behavior such as the Hydra effect [McGrath et al., 2023], whereas a circuit may not. Nonetheless, we suggest that practitioners verify any insights obtained from circuits on the full models wherever possible, regardless of the method used.

## D   How consistent are the circuits found by Edge Pruning?

In this appendix, we evaluate if Edge Pruning can consistently find (i) good circuits, and (ii) consistent circuits in terms of chosen edges across different random initializations. To this end, we choose representative target sparsities ($97.5\%$ for IOI, $99.0\%$ for GT, and $97.0\%$ for GP) and prune a GPT-2 small model with 12 different random seeds with these targets (and other hyperparameters as in Appendix A). As Figures 9 and 10 show, the resulting sparsities and faithfulness of these circuits are remarkably consistent across the 12 seeds, demonstrating that Edge Pruning is robust to different initializations. It is also interesting to ask whether multiple circuits exist for performing a task (and whether Edge Pruning finds them)—Figure 11 investigates this question in this same setting by plotting the distribution of all pairwise IoUs (Intersection-over-Union) in terms of chosen edges, across the $\binom{12}{2} = 66$ pairs of circuits. We observe that the IoU values are generally high (0.5-0.7), but still far from 1. This suggests that while some components may be vital, others might be redundant.

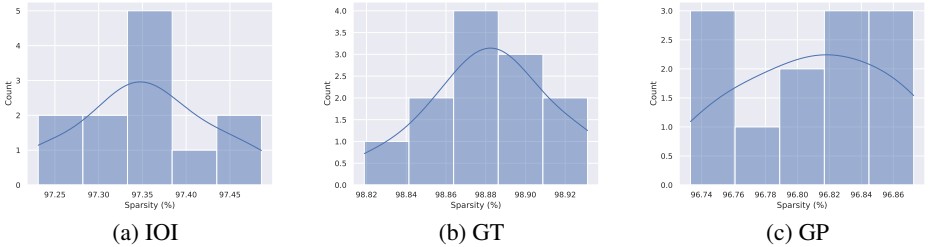

(a) IOI                    (b) GT                    (c) GP

Figure 9: The sparsities of obtained circuits are remarkably consistent across 12 seeds.

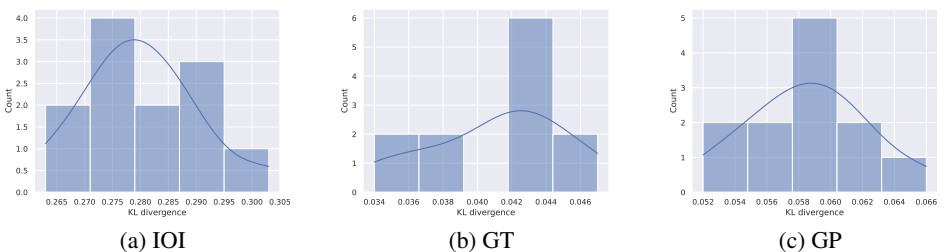

(a) IOI                    (b) GT                    (c) GP

Figure 10: The KL divergences of obtained circuits are consistent across 12 seeds.

This question can be further investigated in future work, especially in how we should define circuits in the face of redundancy.

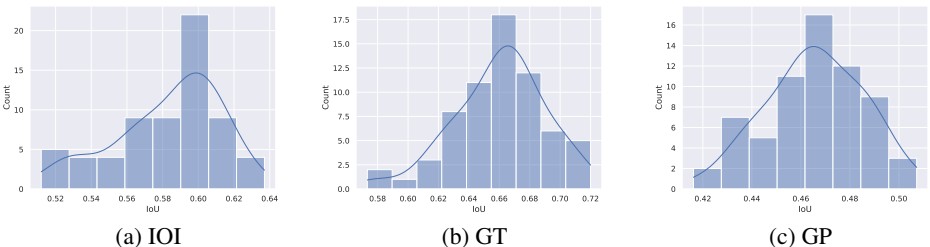

| (a) IOI | (b) GT | (c) GP |

Figure 11: The pairwise Intersection-over-Union over 12 seeds is usually high, but far from 1.

## E    Prompt formats for Boolean Expressions

We show the prompts used for the instruction-prompted and few-shot settings in the CodeLlama-13B case study in Figure 14.

## F    Circuits found with Edge Pruning

In this section, we show example circuit diagrams of the circuits found by Edge Pruning. However, these come with one caveat. Since the typical circuit we found still had too many edges to present in a reasonably sized figure, we only provide figures here for GT and GP, where sparsities ove $99.5\%$ still performed well. Despite this, the circuits here are among the sparsest ones we obtained for each task and therefore perform worse than those at lower sparsities (such as those reported in Figure 2).

The GT circuit is shown in Figure 15, which also reports the faithfulness and performance metrics for it. Similarly, Figure 16 shows a circuit for GP with $99.79\%$ sparsity found by Edge Pruning. Note that the latter, due to the extremely high sparsity, does not perform that well. Nonetheless, the denser circuits compared in prior plots are too unwieldy to show here.

**Interpretation of the CodeLlama-13B circuit.**    Interpreting circuits with $> 1000$ edges remains difficult, but we have made progress in understanding parts of the circuit. For example, we have found the following sub-circuit of two composed heads (refer to Figure 17 for a snippet of this region): L8.H16 attends from operations (and/or) to the previous token (i.e. from op to a in a op b). L10.H24 attends from an operand to a previous operation (i.e. from b to op in a op b) and read the results from L8.H16. This suggests that this duo computes the value of the expression. Interestingly, the attention pattern also holds when a is not a literal like True but an arbitrarily nested subexpression—e.g., attending from or to ( in "((True or False) and True) or False". A hypothesis here is that the model could deal with arbitrary depth expressions by guessing the value of a—allowing it to proceed with the second step—and later verifying the guess. This would also allow the model to parallelize a sequential computation by doing both steps of expression resolution in parallel. Nonetheless, further study and careful interventions are required to verify this hypothesis.

Figure 12: **Intruction prompt**

```
[INST] «SYS»
Evaluate the following boolean
expression as either 'True' or
'False'.
«SYS»

((not not True) and False) or
True [/INST] '
```

Figure 13: **Few-shot prompt**

```
[INST] (True and False) or (False
and True) is [/INST] False
[INST] (True or (not True)) and
False is [/INST] False
[INST] (not (True and False))
or (False and True) is [/INST]
True
((not not True) and False) or True
is [/INST]
```

Figure 14: The prompt used to elicit responses from the CodeLlama-13B model in the instruction prompted and few-shot settings, respectively. The test instance is underlined.

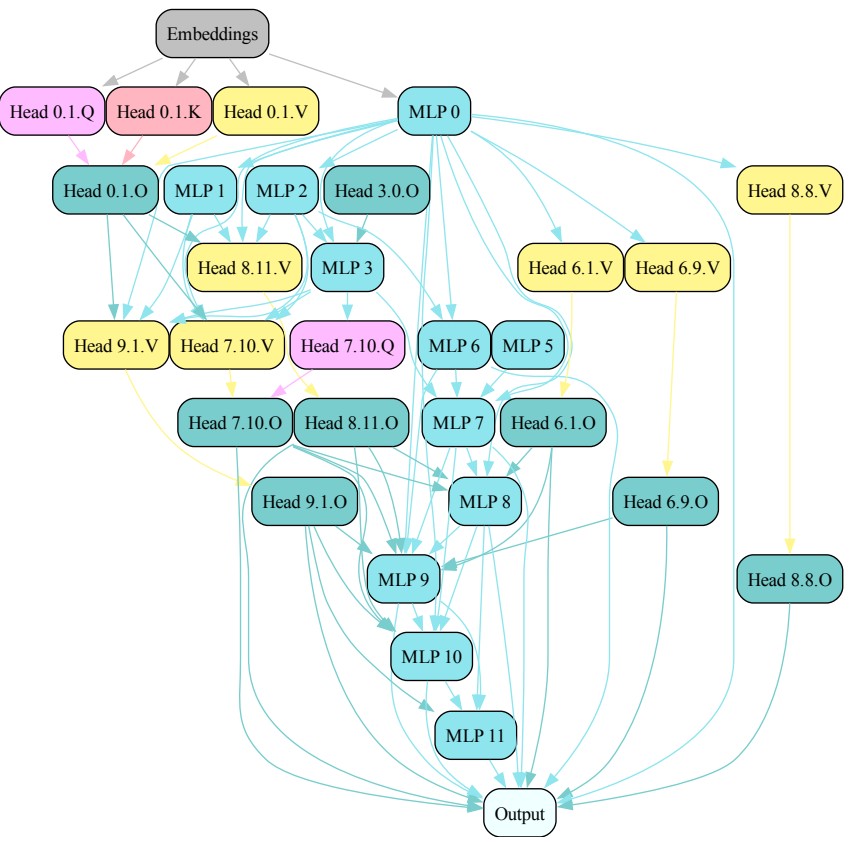

Figure 15: A circuit for GT with 99.77% sparsity, found by Edge Pruning. This circuit obtains a KL divergence of 0.3987 and a Kendall's Tau of 0.7062. The corresponding values for Probability Difference and Probability Difference 10 are 0.4367 and 0.2478, respectively.

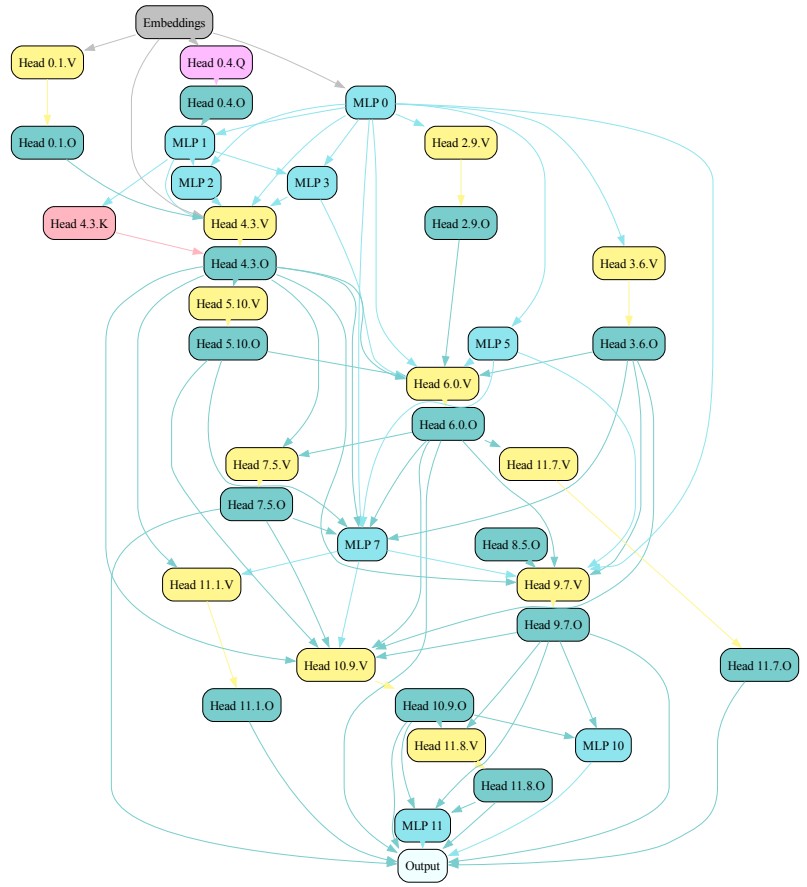

Figure 16: A circuit for GP with $99.79\%$ sparsity, found by Edge Pruning. It obtains a KL divergence of $0.4920$, an accuracy of $55.03\%$, a Logit Difference of $0.9701$, and an Exact Match of $64.02\%$. Note that this circuit does not perform as well as the less sparse ones (see Figure 6). However, we choose to show this circuit here as the denser ones have more edges and are unwieldy to plot.

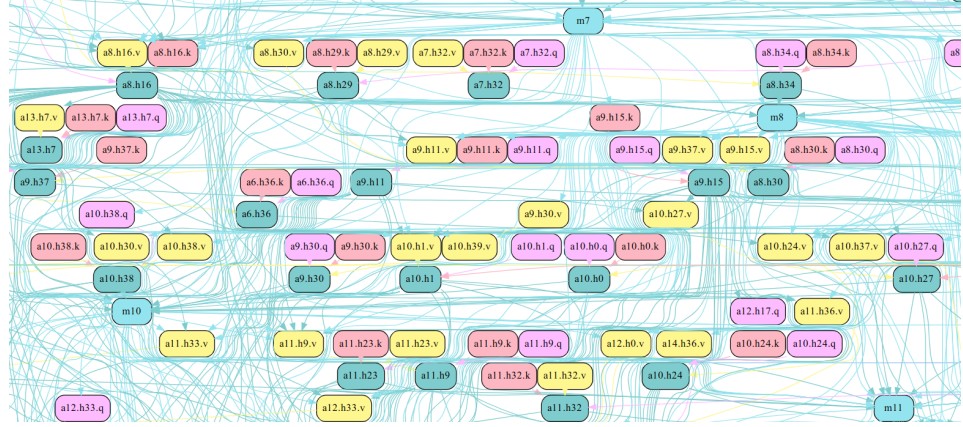

Figure 17: A snippet of the CodeLlama-13B few-shot circuit. The entire circuit is too unwieldy to plot, but this snippet shows a densely connected region. Though a bit hard to make out, `a8.h16` connects to `a10.h24.v`.

