# OpenReview forum: "Finding Transformer Circuits With Edge Pruning"
_NeurIPS.cc/2024/Conference — NeurIPS 2024 spotlight_

### Official Review · Reviewer_TqQ2 · 2024-06-22

**Soundness:** 3
**Presentation:** 3
**Contribution:** 4
**Rating:** 7
**Confidence:** 4

**Summary:**

The paper proposes Edge Pruning, a novel algorithm for circuit finding. They claim that it compares favourably to prior methods on GPT-2 small in terms of circuit metrics like faithfulness. They also claim it scales to the 13B model size. Finally, they apply their model to circuit-finding in a 13B model, and provide preliminary analysis of instruction-following and in-context-learning capabilities.

**Strengths:**

Originality: Excellent. The approach taken to introduce learnable parameters to determine whether to include edges in a circuit is a significant departure from prior work, and opens up exciting new avenues for future research in circuit discovery.

Quality: Good. The experiments conducted are reasonable and the analysis supports the claims made. In particular, it's highly promising that Edge Pruning can recover ground-truth TracR circuits, and outperforms ACDC and EAP by a significant margin on both KL-divergence and logit difference. Some exceptions are discussed in "Weaknesses" below.

Clarity: Fair. The writing was overall clear and flowed well. Specific technical details are not present at time of review, discussed in "Questions" below.

Significance: Excellent. Scalable and effective circuit discovery methods open the door towards accurate interpretability for commercial-sized language models, bringing us closer towards useful applications of interpretability at large.

**Weaknesses:**

1. Circuits are trained to minimize KL divergence, but evaluated using the logit difference. It would be useful to discuss the differences between KL divergence and logit difference, and comment on why they are chosen for training / evaluation respectively.
2. When evaluating baselines, this paper used KL divergence as the objective for both EAP and ACDC. However, both EAP and ACDC originally use logit difference as the optimization objective. Therefore, I am concerned that this is not an apples-to-apples comparison with prior work.
3. The claim that Edge Pruning outperforms EAP / ACDC in GPT-2 small is based primarily on circuit metrics. However, circuit metrics may be misleading. It would be useful to directly compare the nodes and edges found by each method, and report some graph metrics such as node / edge IoU. I would be particularly interested in whether Edge Pruning recovers components of an IOI circuit previously found through manual analysis: https://arxiv.org/abs/2211.00593
4. Similar to the above, there is little / no analysis of circuits found in the 13B model. While it is true that a smaller circuit is generally more interpretable, it is unclear to what extent we can understand the circuits found in the 13B model. It would be useful to highlight specific nodes / edges in the 13B model that seem human-interpretable. As a stretch goal, it would also be useful to formulate and verify hypotheses via standard circuit analysis techniques.

**Questions:**

1. Do you see edge pruning as a suitable technique for unsupervised circuit discovery?
2. Were there other tasks you tried in the CodeLlama-13B model?
3. Other than interchange interventions from a corrupt prompt, did you try other kinds of ablations? (e.g. mean ablation, random-sample ablation, zero ablation)
4. Did circuit-discovery in the 13B model lead you to novel insights about how the model was performing computation?

**Limitations:**

The authors have adequately acknowledged the limitations:
1. Requiring much more GPU memory (32 H100s for Edge Pruning)
2. Being slower than EAP for small dataset sizes

The authors have also acknowledged that circuit metrics may be misleading. It would be good if they could address this limitation as discussed in "Weaknesses".

---

> ### Author Rebuttal · Authors · 2024-08-07
>
> Thank you for your detailed review and feedback. We are happy to find that you found our method original, our experiments convincing and the potential impact high. We respond below to some of the points raised in the review.
>
> > KL divergence v/s Logit Difference for evaluation
>
> We actually evaluate all methods on both KL divergence and Logit Difference, as well as additional metrics. Specifically, we evaluate the methods on KL divergence in Figure 2 and Logit Difference in Figure 3. Additional faithfulness (Exact Match, Figure 5) and accuracy (Figure 6) metrics are provided in Appendix C.
>
> > Training with KL divergence instead of Logit Difference for ACDC and EAP
>
> The ACDC algorithm actually uses KL divergence as the target metric. In Appendix C of the ACDC paper [1], they justify this choice by showing that Logit Difference (or any task-specific metric) can be over-optimized. EAP acknowledges that [1] recommends using KL divergence as the metric as well (Section 3.3 of [2]). However, as 0 is the global optimum for KL divergence, they (EAP) found that the gradient and thus their scores were the zero vector at that point, leading them to use task-specific metrics. It is therefore preferable to use KL divergence as the target when possible, as Edge Pruning does.
>
> > KL divergence v/s circuit overlap metrics
>
> Our choice of evaluation was motivated by [3], which discusses at length why faithfulness metrics are more robust and preferable as evaluation metrics as compared to circuit overlap metrics. In addition, overlap metrics are obtained against manually reverse-engineered circuits, which might not exist yet (e.g. for GP), be incomplete (e.g. IOI – only attention heads were analyzed), not be fine-grained enough (both IOI and GT manual circuits only find important nodes, not edges between them) or even inaccurate (it is difficult to evaluate second order effects like the impact of removing A on the importance of B). Nonetheless, we agree that including these metrics can offer a more rounded comparison. We show the ROC plots for Edge Pruning and ACDC on IOI and GT in the PDF attached with the global response. On the former, Edge Pruning achieves a slightly higher AUC on IOI, and a slightly lower AUC on GT as compared to ACDC. We will also include these results in the next draft.
>
> > Analysis of circuits in the 13B model / Novel insights thereof
>
> Thanks for your suggestion! Please refer to our global response for our additional analysis of the circuit in the 13B model.
>
> **Questions**
>
> > Do you see Edge Pruning as a suitable technique for unsupervised circuit discovery?
>
> We are not quite sure what kind of unsupervised circuit discovery you are referring to in this context. It would be great if you could clarify! In the paper, our method is trained with the KL loss and technically does not need labels during training.
>
> > Did you try other tasks with CodeLlama-13B?
>
> Originally, we also performed experiments with Dyck Languages and Object Counting from BBH with CodeLlama-13B and Llama-2 13B. However, we found that (1) The instruction-prompted performance was usually very low for these tasks, and (2) Llama-2 generally performed worse than CodeLlama on the algorithmic BBH tasks. Hence, we settled upon Boolean Expressions and CodeLlama for the case study.
>
> > Did you try other ablations (mean, zero, random sampling, …)?
>
> That is a good question. We settled on interchange ablations relatively early, and haven’t run the main experiments with other forms of ablations yet. We will include more results in a future version.
>
> [1] “Towards Automated Circuit Discovery for Mechanistic Interpretability”, Conmy et. al., NeurIPS 2023
>
> [2] “Attribution Patching Outperforms Automated Circuit Discovery”, Syed et. al., NeurIPS 2024 ATTRIB Workshop
>
> [3] “Have Faith in Faithfulness: Going Beyond Circuit Overlap When Finding Model Mechanisms”, Hanna et. al., COLM 2024

---

> ### Comment · Reviewer_TqQ2 · 2024-08-12
>
> Thank you for your detailed responses and for including additional metrics, which have addressed most of my concerns with the paper.
>
> I still think it's important to elucidate and analyze an actual circuit using the proposed methodology, as quantitative measures could lead to illusions. (see: https://transformer-circuits.pub/2024/qualitative-essay/index.html). Nonetheless, I understand that circuit analysis is difficult, and based on the empirical evidence available and intuition, my expectation would be quite high that this method leads to interpretable circuits. I strongly recommend the authors to publish the circuit found in Llama-13b in order to aid reproducibility and future work.
>
> Overall, I have decided to keep my current score of 7, recommending acceptance to NeurIPS.

---

### Official Review · Reviewer_qgos · 2024-07-11

**Soundness:** 3
**Presentation:** 2
**Contribution:** 1
**Rating:** 3
**Confidence:** 5

**Summary:**

In this paper, the main focus lies in finding "transformer circuits" to perform a variant of structured pruning on transformers. While generic structured pruning removes neurons, the authors claim that it is a too-coarse approach and propose "edge pruning", where one output wired to more layers can be wired just to a subset of them. The method to achieve this is borrowed from a famous paper in the pruning community [Louizos, 2018]. The empirical results show the effectiveness of the method.

**Strengths:**

- The approach proposed is simple and I can not find a reason why it should not work in general, even beyond transformer networks.

- The presentation is overall clean and fluid, despite a very limited related works section.

**Weaknesses:**

- The authors miss a relevant segment of the structured pruning variant named "channel pruning". Essentially, instead of pruning filters, you can prune input channels to a (convolutional) neural network. Of course, in a feedforward model this results in filter pruning from the previous layer, but already in residual neural networks this is not necessarily true and discovers "circuits". The key factor that allows the proposed approach to work lies in the massive presence of residual branches in modern architectures (in this case, transformers). I reference [A] as one of the most representative works, but a huge effort has been conducted by the pruning community around this topic in the last 6 years. As such, the novelty claims should be massively downscaled.

- Besides, it is unclear what would prevent any practitioner to re-adapt any structured pruning approach along the input dimension. As such, many more comparisons in the experiment section (with either channel pruning approaches or re-adapted structured pruning ones) should be included.

- Furthermore, the proposed approach is very similar to a NAS approach where sub-networks are extracted from a supernet (in your case, the full transformer). A positioning with respect to these methods, and empirical comparison in terms of search complexity and training is also missing.

- Consequently to the previous points, given that the regularizer function to undermine the subnet is borrowed by [Louizos, 2018], I fail to see the technical novelty in this work.

- Besides, across the paper the way the authors address "structured" sparsity is too vague and generic, as well as the fact that they refer to their approach as "Edge Pruning" that should be "different" from unstructured pruning. In reality, unstructured pruning does remove edges (not in the sense of the paper), as each parameter modelizes a connection between the previous and current layer.

[A] He, Yihui, Xiangyu Zhang, and Jian Sun. "Channel pruning for accelerating very deep neural networks." Proceedings of the IEEE international conference on computer vision. 2017.

**Questions:**

The authors are invited to provide a commentary on the differences with the aforementioned literature and to comment on the technical novelty of their work, besides providing answers for the raised points.

**Limitations:**

Limitations are properly discussed.

---

> ### Author Rebuttal · Authors · 2024-08-07
>
> Thank you for pointing us to the literature on Channel Pruning and NAS! We agree this is relevant and will include a discussion of these papers in our next draft. However, we believe there has been a misunderstanding regarding the goal and key contributions of our paper, which we would like to clarify below.
>
> To summarize, our goal is not to develop a novel pruning method. Our main contribution is to *adapt* existing pruning techniques to a problem in transformer interpretability (circuit finding). In particular, to our knowledge, we are the first to prune edges rather than nodes in the computational graph to find circuits, and to scale automated circuit discovery to 10B+ models. As per these goals, the focus of our experiments is in comparing a standard pruning approach to established baseline methods for the circuit finding problem, not in comparing different pruning methods.
>
> Our method is tailored to a particular formulation of “transformer circuits” that has been widely used in the interpretability community [1,2,3,4] and is based on cross-layer edges between interpretable components. As you noted, common pruning techniques produce other kinds of “circuits”, but we highlight below why such circuit definitions have not been as widely adapted in mechanistic interpretability.
>
> > Channel Pruning with residual connections discovers “circuits”... Any practitioner can re-adapt any structured pruning approach along the input dimension.
>
> In the presence of residual connections, pruning the channels of the input would only allow one to either remove *all* the edges from that channel (remove channel) to future layers or *none* of them (keep channel). Edge Pruning isolates the exact contribution between any two layers, which gives interpretability practitioners a much more precise signal for understanding the internal control flow of the model.
>
> Another difficulty with pruning the channel or hidden size dimension is that the resulting subspaces are difficult to interpret and suffer from “polysemanticity” [5]. Therefore, interpretability research tends to focus on larger components such as attention heads and retain all hidden dimensions, which can be made more interpretable with tools such as LogitLens [6, 7]. In the common response, we discuss how Edge Pruning could be extended to prune more semantic features in the hidden states of the model.
>
> > Given that the regularizer function to undermine the subnet is borrowed by [Louizos, 2018], I fail to see the technical novelty in this work.
>
> By adapting pruning techniques to the circuit finding problem, we contribute several technical changes to prior methods. These include defining masks for edges between components across layers; disentangling the residual stream from each component to each downstream component; and replacing pruned components with counterfactual activations, rather than 0's. Note that these contributions would not make sense in the context of the standard pruning literature.
>
> > [The authors] refer to their approach as "Edge Pruning" that should be "different" from unstructured pruning. Unstructured pruning does remove edges (not in the sense of the paper), as each parameter modelizes a connection between the previous and current layer
>
> We can see how the name “edge pruning” could technically apply to many well-known techniques commonly referred to as “unstructured pruning”. However, we hope that our method “Edge Pruning” will be primarily associated with transformer interpretability, and therefore not cause any confusion in the pruning/model compression community.
>
>
> [1] “Interpretability in the Wild: a Circuit for Indirect Object Identification in GPT-2 Small”, Wang et. al., ICLR 2023
>
> [2] “How does GPT-2 compute greater-than?: Interpreting mathematical abilities in a pre-trained language model”, Hanna et al., NeurIPS 2023
>
> [3]  “Towards Automated Circuit Discovery for Mechanistic Interpretability”, Conmy et. al., NeurIPS 2023
>
> [4] “Circuit Component Reuse Across Tasks in Transformer Language Models”, Merullo et. al., ICLR 2024
>
> [5] “Feature visualization”, Chris Olah, Alexander Mordvintsev, and Ludwig Schubert, Distill 2017
>
> [6] “Interpreting GPT: The logit lens”, nostalgebraist, LessWrong 2020
>
> [7] “Eliciting Latent Predictions from Transformers with the Tuned Lens”, Belrose et al., 2023

---

> > ### Comment · Reviewer_qgos · 2024-08-12
> >
> > After reading the other reviewer's comments, and the author's rebuttal, I still feel that this work has very limited technical novelty (also including the further details provided, which can be simply summarized as codework + replacement with counterfactual activations, that are also borrowed from another work!).
> >
> > I disagree with the authors when they say that structured pruning "either removes all the edges from that channel (remove channel) to future layers or none of them (keep channel)" - it just depends on whether it is applied at the output of a layer (in that case, the authors are right) or at the *input* of a layer (in that case, you would obtain *exactly the same effect as observed here*). A comparison with these approaches would be a must.
> >
> > In general, I also find the claims about interpretability, as the same authors declare, difficult to handle, due to the largeness of the models under exam.
> >
> > All of the issues are in my opinion unaddressable at this stage. Although raising my score, my evaluation is still a reject.

---

> > > ### Author Response · Authors · 2024-08-13
> > > **Author Response**
> > >
> > > Thank you for responding!
> > >
> > > > Comparison with [structure pruning] is a must
> > >
> > > The suggested baseline of applying structured pruning to the inputs of a layer is part of our proposed method, which also separates the contribution from each previous layer. We re-iterate that pruning individual neurons/channels would not be a useful comparison, since the resulting circuits would not be desirable for mechanistic analysis of the model due to the sheer volume and polysemanticity of neurons, and lack of meaningful "counterfactual" channel features.
> > >
> > > > I also find the claims about interpretability, as the same authors declare, difficult to handle, due to the largeness of the models under exam.
> > >
> > > It seems to us that this is rather a judgement on the whole research area than the merit of our work and, as such, would ask the reviewer to re-consider the confidence score of their review. We point to the published and widely cited AC/DC [1] - a direct predecessor to our method which produces circuits of the same granularity - to argue that automatic circuit finding is indeed a useful and promising approach in transformer interpretability.
> > >
> > > [1] "Towards Automated Circuit Discovery for Mechanistic Interpretability", Conmy et. al., NeurIPS 2023

---

### Official Review · Reviewer_HMkg · 2024-07-11

**Soundness:** 3
**Presentation:** 3
**Contribution:** 4
**Rating:** 8
**Confidence:** 3

**Summary:**

The authors propose "Edge Pruning" as an effective and scalable method for automatic circuit discovery. Edge Pruning consists of learning a binary mask over the edges of the computation graph in a transformer neural network. Edge pruning performs favorably compared to the prior art and the authors demonstrate how it can successfully be used to find new circuits in a case study.

**Strengths:**

- The proposed Edge Pruning method is novel and a worthy contribution to the literature.
- The comparisons to the prior are are thorough.
- The paper is well-written and well-organized. The details of the Edge Pruning method are explained well.
- The case study in which the authors find a circuit in CodeLlama-13B provides excellent evidence of the utility of Edge Pruning. The circuit that they uncovered will be able to be used as an extra "ground truth" circuit that all future work can use.

**Weaknesses:**

- It would have been useful to see whether or not any prior methods for "automatic" circuit discovery could have found the same or a similar circuit in CodeLlama on the same task.

**Questions:**

- Did you attempt to use any other circuit discovery methods on the CodeLlama task?
- You acknowledge in the limitations that it is difficult for a human to fully interpret these computational graphs with hundreds of edges, but (and this is not meant as a criticism) how hard have you tried?
- Could Edge Pruning scale to neuron or subspace-level components?

**Limitations:**

The outputs of circuit discovery techniques such as Edge Pruning are large computation graphs that are not, in and of themselves, interpretable.

---

> ### Author Rebuttal · Authors · 2024-08-07
>
> Thank you for leaving a thoughtful and constructive review. It endears us to learn that you found our method novel, and our case study a good demonstration of its utility. We take this opportunity to respond to some of your questions here.
>
> > Other methods on CodeLlama
>
> We couldn’t run ACDC on CodeLlama as it was prohibitively expensive. Although EAP can work with CodeLlama in theory, we struggled to run it as its current implementation does not interface easily with scalable frameworks like FSDP/DeepSpeed. We expect that an appropriate implementation of EAP will remain efficient for CodeLlama 13B.
>
> > Human interpretation
>
> While it is challenging to reverse-engineer a large circuit, we believe parts of it can still be understood with enough effort. We discuss this point further in the global response.
>
> > Can Edge Pruning scale to neuron or subspace-level components?
>
> Thanks for the question! Please refer to the global response for how Edge Pruning may be used in this setting.

---

> > ### Comment · Reviewer_HMkg · 2024-08-13
> >
> > Thank you for your responses. I remain excited about this submission and am keeping my score.

---

### Official Review · Reviewer_xxo6 · 2024-07-12

**Soundness:** 3
**Presentation:** 3
**Contribution:** 4
**Rating:** 8
**Confidence:** 4

**Summary:**

The paper proposes a method to automatically discover circuits within trained transformer models that are responsible for its behavior on a specific task. The automatic discovery of those circuits forms the first step to interpret the trained model. The transformer is visualized as a graph with each attention head containing 4 nodes, 3 nodes for key, query and value producing neurons with incoming edges and 1 node for the output with outgoing edges. For a specific task, the optimization problem is formulated as the difference between the output of the full model against the output of the circuit given original and corrupted inputs. The use of corrupted inputs act as a means to identify and suppress network components that are not important for the eventual circuit output.

Previous methods include greedy search and linear approximation-based edge scoring for circuit discovery. The authors propose an edge pruning based method where each edge is assigned a binary mask value relaxed to be continuous for optimization. This mask value for a specific edge is applied to the original input while the corrupted input receives the compliment and then passed on to the node. The loss used for the output is KL divergence between the circuit output and the output of the full model. After optimizing, the continuous mask values are converted into binary values by simple thresholding. Thus, a sparse graph within the overall graph is discovered that faithfully captures the behavior of the full model on the specific task.

The authors evaluate the proposed method on a wide range of tasks and metrics. The results show that the circuits uncovered are more faithful to the full model and perform better as compared to circuits discovered through previous methods. Additionally, the circuits are far sparser as compared to the previous methods. Finally, the authors show that the method can be scaled to a 13 Billion parameter model showcasing the applicability of the method.

**Strengths:**

1. The paper proposes an interesting and efficient method on an important problem in transformer interpretability.
2. The experiments conducted and results shown are convincing of the methods superiority in terms of faithfulness and scalability.

**Weaknesses:**

1. The proposed method and direction are useful for discovering circuits for specific tasks. However, components of the circuits uncovered may have varied behavior across tasks or even input datapoints not considered in the dataset. This makes the overall goal of mechanistically understanding each component dependent on the specific task and the corresponding data available.
2. The use of KL divergence is prominent in the literature. Do you also consider the possibility that multiple circuits may emerge in the model with similar output behavior? It is unclear whether the discovered circuits are consistent across models trained using different seeds.
3. One of the major concerns is that the circuit is selected based on the sparsity level. Multiple circuits can be selected from the same optimization run depending on the sparsity and faithfulness. How do the authors decide the final circuit in terms of sparsity to be used for the next step of reverse engineering?

**Questions:**

The experiments conducted show that circuits discovered are faithful and sparse. However, the authors should also consider reverse engineering the circuits uncovered to showcase that the circuits are not only faithful to the overall model behavior but also interpretable.

The circuits are discovered on a component level and not on a neuron level would interpreting the resulting circuits be a time-consuming avenue. How would the method work on a weight level mask ?

How do the learning rates for different component masks affect the final circuit? It is unclear what would be a good choice for the hyperparameters.

How are the mask values initialized ?

**Limitations:**

The authors discuss the limitations of their method in a sufficient level of detail.

---

> ### Author Rebuttal · Authors · 2024-08-07
>
> Thank you for providing valuable feedback and suggestions. We are happy to read that you found our method interesting and useful, and the methods convincing. We would like to respond below to some of the questions and concerns raised.
>
> > Circuits may have varied behavior on tasks or input datapoints not considered in the dataset
>
> Circuit finding is indeed sensitive to the task and which datapoints are selected. Our formulation of circuit finding as loss minimization makes this dependence explicit and we evaluate the “in-distribution” generalization of a circuit to unseen examples from the same task distribution. It would be interesting for future work to explore more intuitive ways of defining tasks and evaluate the “out-of-distribution” behavior of circuits, e.g., on task compositions.
>
> > Could there be multiple circuits with a similar KL?
>
> Excellent question!  We explore this by running Edge Pruning with 12 different seeds and a fixed sparsity target across three tasks and measure their pairwise similarity via Intersection-over-Union (IoU). Please refer to the global response, which shows that the KL values are consistent across runs. However, the IoU values are in the range of 0.52—0.64, indicating the existence of many partially overlapping circuits with similar KL. We argue that this does not represent a weakness of our method, as all these circuits are valid solutions to the problem statement. However, an analysis of their commonalities and differences could motivate future work and we will include this discussion in the next draft.
>
> > How to select the sparsity level to be used for the next steps of reverse engineering?
>
> In circuit finding, there is a trade-off between circuit sparsity and faithfulness to the full model: larger circuits are harder to interpret, but more faithful to the full model. One option is to find circuits with a range of target sparsities and identify inflection points in the KL-sparsity curve.
>
> **Questions**
> > Interpreting some of the circuits discovered
>
> Great suggestion! We plan to provide a detailed analysis in the next draft of the paper. We outline our ongoing efforts in the global response.
>
> > Component v/s neuron level
>
> We discuss the challenges of extending Edge Pruning to neural circuits in the global response.
>
> > Learning rates
>
> Our method requires a high learning rate. Our experiments use a learning rate of 0.8 for all masks (log alphas) and all regularization lambdas, which we found to work well across many batch sizes and training steps.
>
> > Initialization of masks
>
> Following [1], we initialize our log-alphas to a normally distributed tensor with each entry having a mean of 10.0 and a standard deviation of 0.01 (corresponding to initial masks close to 1). The regularization lambdas start at 0.
>
> [1] “Structured Pruning Learns Compact and Accurate Models”, Xia et. al., ACL 2022

---

> ### Comment · Reviewer_xxo6 · 2024-08-10
>
> thanks for the responses to my reviews. I have also read the comments by the other reviewers. I agree with some of the points made by the more negative reviewer (qgos) but I think those can be addressed with writing/presentation changes.

---

### Author Rebuttal · Authors · 2024-08-07

We thank the reviewers for their insightful comments. We are happy to see that the reviewers found our method interesting, the experiments convincing, and the potential impact high.
At the same time, two questions were raised by multiple reviewers. We would like to address them here.

> Can Edge Pruning work with neurons?

Individual neurons are typically polysemantic, i.e., [1] show how a single neuron responds to academic citations, English dialogue, HTTP requests and Korean text. Recent work suggests that Sparse Auto-Encoders (SAEs) trained to reconstruct the hidden states of a language model learn sparse features that correlate with distinct, interpretable concepts [2]. Recently, [3] have adapted EAP to work with “feature circuits” over such SAE features rather than heads/MLPs. The number of all feature edges is prohibitively high (quadratic in #features), and [3] used a two-stage approach where they first found important features (nodes), and then found important edges between them. Edge Pruning could be similarly adapted in this fashion: we can first model masks over features to find important features, and then learn masks over pairs of them. Since the former step only involves as many masks as the number of features, its memory overhead should be a constant factor.

> How hard is it to interpret the features of the 13B model / Did you find anything interesting?

Interpreting circuits with >1000 edges remains difficult, but we have made progress understanding parts of the circuit. For example, we have found the following circuit of two composed heads:
L8.H16 attends from operations (and/or) to the previous token (i.e. from op to a in “a op b”)
L10.H24 attends from an operand to a previous operation (i.e. from b to op in “a op b”) and read the results from L8.H16
This suggests that this duo computes the value of the expression. Interestingly, the attention pattern also holds when a is not a literal like “True” but an arbitrarily nested subexpression. A hypothesis here is that the model could deal with arbitrary depth expressions by guessing the value of “a”—allowing it to proceed with the second step—and later verifying the guess. This would also allow the model to parallelize a sequential computation by doing them in parallel. Nonetheless, further study and careful interventions are required to verify this hypothesis. We will include these preliminary findings in the next draft and hope that it may inspire future work to study these circuits in more detail.

Finally, we have attached a PDF with this common response containing figures that we referred to in some of the responses. The first figure runs Edge Pruning with multiple seeds and finds that we discover multiple partially overlapping circuits with similar (good) faithfulness at the same sparsity. The second demonstrates that Edge Pruning is competitive on circuit overlap metrics when compared to ACDC.

[1] “Towards Monosemanticity: Decomposing Language Models With Dictionary Learning”, Bricken et al., https://transformer-circuits.pub/2023/monosemantic-features

[2] “Sparse Autoencoders Find Highly Interpretable Features in Language Models”, Cunningham et. al., ICLR 2024

[3] “Sparse Feature Circuits: Discovering and Editing Interpretable Causal Graphs in Language Models”, Marks et. al., arXiv 2024

---

### Decision · Program_Chairs · 2024-09-25

**Decision:**

Accept (spotlight)

**Comment:**

This paper proposes a novel automated method to discover circuits in transformer models. The method uses a technique from the pruning literature, adapted to this task. The paper gives good comparison with literature and is well written. The experiments show that the method scales to large models and is an important contribution in the field of interpretability of large language models.